# Ecotype-Specific Pathways of Reactive Oxygen Species Deactivation in Facultative Metallophyte *Silene vulgaris* (Moench) Garcke Treated with Heavy Metals

**DOI:** 10.3390/antiox9020102

**Published:** 2020-01-24

**Authors:** Ewa Muszyńska, Mateusz Labudda, Adam Kral

**Affiliations:** 1Department of Botany, Institute of Biology, Warsaw University of Life Sciences-SGGW, Nowoursynowska 159, Building 37, 02-776 Warsaw, Poland; adam.stefan.kral@gmail.com; 2Department of Biochemistry and Microbiology, Institute of Biology, Warsaw University of Life Sciences-SGGW, Nowoursynowska 159, Building 37, 02-776 Warsaw, Poland; mateusz_labudda@sggw.pl

**Keywords:** antioxidants, bladder campion, hormesis, metallic elements, pigments, ultrastructure

## Abstract

This research aimed to indicate mechanisms involved in protection against the imbalanced generation of reactive oxygen species (ROS) during heavy metals (HMs) exposition of *Silene vulgaris* ecotypes with different levels of metal tolerance. Specimens of non-metallicolous (NM), calamine (CAL), and serpentine (SER) ecotypes were treated in vitro with Zn, Pb, and Cd ions applied simultaneously in concentrations that reflected their contents in natural habitats of the CAL ecotype (1× HMs) and 2.5- or 5.0-times higher than the first one. Our findings confirmed the sensitivity of the NM ecotype and revealed that the SER ecotype was not fully adapted to the HM mixture, since intensified lipid peroxidation, ultrastructural alternations, and decline in photosynthetic pigments’ content were ascertained under HM treatment. These changes resulted from insufficient antioxidant defense mechanisms based only on ascorbate peroxidase (APX) activity assisted (depending on HMs concentration) by glutathione-*S*-transferase (GST) and peroxidase activity at pH 6.8 in the NM ecotype or by GST and guaiacol-type peroxidase in the SER one. In turn, CAL specimens showed a hormetic reaction to 1× HMs, which manifested by both increased accumulation of pigments and most non-enzymatic antioxidants and enhanced activity of catalase and enzymes from the peroxidase family (with the exception of APX). Interestingly, no changes in superoxide dismutase activity were noticed in metallicolous ecotypes. To sum up, the ROS scavenging pathways in *S. vulgaris* relied on antioxidants specific to the respective ecotypes, however the synthesis of polyphenols was proved to be a universal reaction to HMs.

## 1. Introduction

Plants, as sessile organisms, are frequently exposed to a wide array of hostile conditions, which stimulate them to adjust their metabolism to external environmental stressors, including salinity, drought, and heavy metal contamination. The last-mentioned factor seems to be particularly alarming, since the incidence of heavy metals (HMs) in the biosphere has been grown hazardously over the last several decades due to the ongoing global industrialization and urbanization [1]. Some HMs, such as zinc (Zn), nickel (Ni), or copper (Cu), are fundamental to different physiological processes and required in trace amounts for higher plants [2]. Others, like lead (Pb) and cadmium (Cd), do not have recognized beneficial roles, although more and more research has recently indicated their positive effects on morpho-physiological traits under low-dose treatment [3,4,5,6]. Despite their biological necessity, HMs become toxic if their concentration exceeds the genotype-dependent thresholds. What is more, all metals are non-biodegradable, and therefore they persist inside the protoplast, where they may cause cellular damage and interfere with various physiological processes [7,8,9]. 

The primary effect of HM exposition is the overproduction of reactive oxygen species (ROS). Although ROS play an integral role in the regulation of plant metabolism, their enhanced formation induces oxidative damage, and leads to the oxidation of lipids, proteins, or nucleic acids and the depletion of enzymatic activities [10,11,12,13]. However, plants activate various defense mechanisms to overcome the destructive oxidative reactions and protect the cellular components from imbalanced ROS generation under HM stress. The antioxidant machinery comprises of enzymatic and non-enzymatic systems. The first group of antioxidants includes, among others, superoxide dismutase (SOD), catalase (CAT), and peroxidase (POX), while the second—phenols, flavonoids, proline, and carotenoids [14]. All these compounds possess the ability to inactivate ROS in a complex way by their decomposition or transformation into less harmful molecules [15]. What is more, phenolic compounds show an additional tendency to chelate metallic ions through hydroxyl and carboxyl groups, as well as participate in pathways of cell wall lignification, causing the formation of a thicker mechanical barrier that restricts the penetration of HMs responsible for oxidative damage [7,9,16]. Therefore, modulation of the particular antioxidants production and accumulation depending on the stressor presence can be used to anticipate the direction of metabolism remodeling under HM treatment. 

The toxicity of HMs applied separately has been intensively studied in taxonomically diversified organisms [8,17,18]. However, the impact of co-treatment with different ions on cellular homeostatic pathways still remains to be elucidated. It seems to be a particularly interesting issue if genotypes representing ecologically distinct environments are taken into consideration. Such specialized genotypes may spontaneously appear on chemically degraded areas, which provoke natural selection, leading to the development of adaptive traits to elevated concentrations of HMs [19,20,21]. These ecotypes of commonly known species, called facultative metallophytes, are able to survive in the presence of an extremely high level of metallic elements without suffering any phytotoxic effects. On the contrary, when their non-adapted forms from unpolluted terrains face prolonged HM stress or excessive amounts of toxic substances, their defense responses may be exhausted. One of the facultative metallophytes is the bladder campion (*Silene vularis*) belonging to the *Caryophyllaceae* family. Our recent study showed that contrasting ecotypes of this species from two metal-tolerant populations growing on waste heaps deposited after the mining and processing of calamine and serpentine rocks differed significantly in anatomical, physiological, and genetic features, not only from reference specimens occurring in unpolluted areas, but to some extent also to each other [22]. What is more, under fully controlled in vitro conditions, we have ascertained the survival strategy resulting from the ecotype-dependent activation of antioxidant machinery in these three ecotypes treated separately with Pb or Ni ions, i.e., metals which are characteristic for natural habitats of metallicolous populations [9]. Taken together, these experiments revealed that ROS may play an essential function as signaling molecules in response of metallicolous specimens to HMs. This prompted us to establish a new experimental scheme in order to learn more about the antioxidant defense system of *S. vulgaris*. In the present study, we evaluate the specific responses of non-metallicolous, calamine, and serpentine ecotypes to simultaneous treatment only with metals characteristic of the calamine substrate (Zn, Pb, and Cd), but applied at various concentrations. Such comparative analyses to the merely one group of metals in related specimens with a divergent level of tolerance to different ions can provide useful information not only about strategies involved in ROS scavenging and thus metal tolerance *sensu lato*, but also about ecotype-specific and cross reactions to HMs. Therefore, the aim of this study was: (i) to evaluate the efficiency of enzymatic and non-enzymatic antioxidant machinery under HM stress, (ii) to demonstrate specific and unspecific antioxidant responses in particular ecotypes, (iii) to indicate which antioxidant pathways are exclusive and/or activated in a certain metal toxicity. To achieve the set goals, the experimental model was based on shoot culture, allowing the exclusion of the probability of metal immobilization in roots. 

## 2. Materials and Methods

### 2.1. Plant Material and Experimental Scheme

Plant material constituted the in vitro shoot culture of three ecotypes of *Silene vulgaris* (Moench) Garcke (*Caryophyllaceae*) that were propagated on the optimized proliferation medium according to the protocol proposed by Muszyńska et al. [22]. Two HM-tolerant cultures were obtained earlier from seeds of specimens representing calamine (described further as CAL) and serpentine (SER) populations, while control non-metallicolous (NM) culture originated from seeds collected from an unpolluted stand [9,22]. In the experimental period, apical fragments of shoots (explants) were placed on the proliferation medium enriched simultaneously with zinc, lead, and cadmium salts. In the first treatment, the applied doses of HMs reflected their contents in the natural habitat of the calamine ecotype, i.e., 714.3 μM ZnSO_4_, 3.0 μM Pb(NO_3_)_2_, and 16.4 μM CdCl_2_ (described further as 1× HMs) [22]. In the following, HM concentration was 2.5- and 5.0-times higher than the first one (described further as 2.5× HMs and 5× HMs, respectively). Control cultures were multiplicated on the metal-free medium. Five microcuttings per 200 mL flask were put on 50 mL of the respective culture medium. Six flasks (replicates) per each treatment were applied. The experiment lasted eight weeks, with subculture after four weeks, when explants were transferred onto freshly prepared media with the same composition. The exception was the SER culture cultivated on medium enriched with 2.5× HMs, for which dying was observed after six weeks of cultivation. Therefore, the physiological and anatomical studies for this treatment were performed earlier, when cultures were still vigorous, in order to find the reasons for such far-reaching changes. 

### 2.2. Growth Parameters’ Determination

The rate of culture growth was evaluated on the basis of the multiplication coefficient (MC), which was calculated after four and eight weeks of treatment as a ratio of newly regenerated adventitious shoot number per primary explant. Additionally, shoots were measured after eight weeks. Then, plant material was weighted and dried at 105 °C to a constant mass in order to determine fresh (FW) and dry (DW) matter content, respectively. 

### 2.3. Examination of Leaf Anatomy and Ultrastructure 

Six leaf blade fragments (approximately 3 × 3 mm in size, taken from the central leaf part) from each treatment were primary fixed in 2% glutaraldehyde and 2% paraformaldehyde dissolved in 100 mM cacodylate buffer (pH 7.2) for 3 h. After four rinsing series in 100 mM cacodylate buffer, samples were post-fixed in 2% OsO_4_ (for 2 h in the refrigerator), dehydrated in increasing ethanol concentrations, and finally embedded in medium epoxy resin (EPON 812). The curing of prepared blocks was performed at 60 °C for 24 h. Then, blocks were cut using microtomes (Jung RM 2065 and UCT ultra microtome, Leica Microsystems) on semi-thin (3 μm in thick) and ultra-thin (90 nm thick) sections. Semi-thin sections were stained with methylene blue and azure B for examination under a light microscope (Olympus-Provis), while ultra-thin sections were stained with uranyl acetate followed by lead citrate for observation under transmission electron microscope (FEI 268D ‘Morgagni’ FEI Company, Hillsboro, OR, USA). 

### 2.4. Evaluation of Photosynthetic Pigment Content

Photosynthetic pigment content was determined according to Lichtenthaler [23]. One hundred milligrams of leaf samples were homogenized with 80% acetone in ice-cold conditions and centrifuged. The absorbance of pigments was measured at 646 and 663 nm with BioSpectrometer kinetic (Eppendorf, Hamburg, Germany). The chlorophyll *a* (chl *a*) and chlorophyll *b* (chl *b*) contents were calculated according to Wellburn [24] equations and expressed as mg g^−1^ FW of the sample. Additionally, the ratios of chl *a* to chl *b* as well as total chlorophylls (chl *a+b*) to carotenoids (car) were counted. 

### 2.5. Determination of Lipid Peroxidation 

Malondialdehyde (MDA) content, reflecting lipid peroxidation, was measured after the homogenization of 100 mg of plant material with 80% methanol. Samples were centrifuged (4 °C, 16,000× *g*, 20 min) and 200 μL of methanolic extract was added to 800 μL 0.5% 2-thiobarbituric acid dissolved in 20% trichloroacetic acid solution. After incubation at 90 °C for 20 min, reactions were stopped on an ice bath. Samples were centrifuged (16,000× *g*, 10 min) and the supernatant’s absorbance was measured at 440 nm, 532 nm, and 600 nm in Nunc U-bottom 96-well plates (Thermo Scientific, Waltham, MA, USA) on a Varioskan LUX Multimode Microplate Reader (Thermo Scientific, Waltham, MA, USA). MDA level was calculated according to Hodges et al. [25] and expressed in µmol per gram FW. 

### 2.6. Assessment of Antioxidative System Efficiency

#### 2.6.1. Non-Enzymatic Antioxidants

##### Carotenoids

Carotenoids (car) extraction from plant tissue was performed with 80% acetone. The absorbance of centrifuged extracts was measured spectrophotometrically with BioSpectrometer kinetic and Wellburn’s equation [24] was used to calculate carotenoids content:car = (1000·A_470_ − 3.27·chl *a* content − 104·chl *b* content)/198.

##### Proline

Free proline concentration was extracted with 3% aqueous sulfosalicylic acid from 100 mg of plant tissue and estimated with ninhydrin reagent [26]. The reaction was conducted for 1 h at 100 °C and terminated on ice. After cooling to room temperature (RT), solutions were mixed with toluene. The absorbance of toluene fractions was read at 520 nm with BioSpectrometer kinetic. Proline level was estimated using calibration curve and expressed as mg of proline per 100 g FW.

##### Phenolic Profile

Phenolic compounds were estimated after the homogenization of plant material (100 mg) with 80% methanol. The polyphenol content was determined according to Swain and Hillis’s [27] method. Centrifuged extracts were mixed with deionized water, Folin–Ciocalteu reagent, and saturated Na_2_CO_3_. The mixtures were kept at 40 °C for 30 min, and after cooling to RT the absorbance at 740 nm was read. Polyphenols were expressed as mg gallic acid equivalent per 100 g of FW. For the determination of phenylpropanoids, flavonols, and anthocyanins, the Fukumoto and Mazza method was applied [28]. Methanolic extracts were mixed with 0.1% HCl (in 96% ethanol) and 2% HCl (in water). The absorbances at 320 nm, 360 nm, and 520 nm were determined with BioSpectrometer kinetic after 15 min. Caffeic acid, quercetin, and cyanidin were used as standards for the measurement of phenylpropanoids, flavonols, and anthocyanin contents, respectively. The amount of particular phenols was expressed in mg per 100 g FW. Additionally, the sum of compounds with double bonds showing maximum absorbance at 280 nm was estimated. They were classified as the total secondary metabolites and expressed as mg of chlorogenic acid per 100 g FW. 

#### 2.6.2. Antioxidant Enzymes’ Assay

One hundred mg of *S. vulgaris* shoots was macerated in mortar with quartz sand and 1 mL of ice-cold extraction medium (pH 7.2) (50 mM 3-(N-morpholino)propanesulfonic acid, 2 mM 2-mercaptoethanol, 0.1 mM ethylenediaminetetraacetic acid, 2% polyvinylpyrrolidone, 0.5% Triton X-100, 1 mM phenylmethylsulfonyl fluoride). After incubation on ice for 20 min, samples were centrifuged (4 °C, 16,000× *g*, 20 min). The extracts were stored at −80 °C until analysis. 

##### Superoxide Dismutase (SOD)

SOD activity was measured using Kostyuk and Potapovich’s [29] method. A reaction medium was obtained by mixing equal volumes of 67 mM K/Na phosphate buffer (pH 7.8) with 25 mM ethylenediaminetetraacetic acid (EDTA). The pH value of this solution was adjusted to 10.0 by tetramethylethylenediamine. Then, 50 μL of reaction medium and 120 μL Milli-Q water were added to 5 μL of extract. Five μL of 2.5 μM quercetin in dimethyl sulfoxide was used to initiate the reaction. Assays were conducted in Nunc U-bottom 96-well plates on a Varioskan LUX Multimode Microplate Reader. The absorbance at 406 nm was recorded for 20 min with absorbance reads every 1 min. The activity of SOD was shown in arbitrary units (the amount of SOD that inhibits superoxide-induced quercetin oxidation by 50%) per gram of FW. 

##### Catalase (CAT)

CAT activity was measured according to Aebi [30]. Two μL of extract was mixed with 18 μL of 50 mM Tris-HCl buffer (pH 7.0) and 10 μL of 0.168% H_2_O_2_ in the same buffer. Assays were conducted at 37 °C in UV-Star 96-well plates (Greiner, Monroe, NC, USA) on a Varioskan LUX Multimode Microplate Reader. The absorbance at 240 nm was recorded for 10 min with absorbance reads every 30 s. The CAT activity was expressed as a decomposition of µmol of H_2_O_2_ per minute and gram FW. 

##### Peroxidase Family (POX)

Guaiacol-type peroxidase (GOPX) was measured using Chance and Maehly’s [31] method. The extract (5 μL) was mixed with 5 mM guaiacol and 2.5 mM H_2_O_2_ in 50 mM acetic buffer (pH 5.6). Assays were conducted at 37 °C in Nunc U-bottom 96-well plates on a Varioskan LUX Multimode Microplate Reader. The absorbance at 470 nm was recorded for 10 min with absorbance reads every 30 s. The GOPX activity was expressed in µmol of formed tetraguaiacol (ε  =  26.6 mM^−1^ cm^−1^) per minute and gram FW. 

Peroxidase activity (POD) was measured according to Lück [32]. The extract (5 μL) was mixed with an assay medium (0.49% *p*-phenylenediamine (PPD) and 0.049% H_2_O_2_ in 50 mM Tris-HCl buffer, pH 6.8 or 8.8). Assays were conducted at 37 °C in Nunc U-bottom 96-well plates on a Varioskan LUX Multimode Microplate Reader. The absorbance at 485 nm was recorded for 10 min with absorbance reads every 30 s. The oxidation of PPD by POD/H_2_O_2_ caused the production of Bandrowski’s base and the POD activity was expressed in arbitrary units, separately for pH 6.8 (as POD_6.8_) and 8.8 (as POD_8.8_). The one unit of POD activity was regarded as a 0.1 increase in absorbance after 1 min per gram FW. 

Ascorbate peroxidase (APX) activity was ascertained using Nakano and Asada’s [33] method. The extract (10 μL) was mixed with 50 mM Tris-HCl buffer (pH 6.8), 2 mM sodium ascorbate, 5 mM EDTA, and 0.1 mM H_2_O_2_. The APX activity was monitored at 25 °C in UV-Star 96-well plates on a Varioskan LUX Multimode Microplate Reader by measuring the rate of ascorbate degradation for 10 min with absorbance reads every 30 s at 290 nm. The APX activity was expressed in µmol of ascorbate (ɛ  =  2.8 mM^−1^ cm^−1^) per minute and gram FW. 

##### Glutathione-*S*-Transferase (GST)

GST activity was measured using the modified method by Habig et al. [34]. The activity assay medium consisted of 0.1 M Tris-HCl buffer (pH 6.8), 5 mM reduced glutathione (GSH), 1 mM of 1-chloro-2,4-dinitrobenzene (CDNB), 1% ethanol, and 50 μL of extract in a total volume of 141.2 μL. Reactions were carried out at 37 °C in Nunc U-bottom 96-well plates on a Varioskan LUX Multimode Microplate Reader and the change in absorbance at 340 nm was monitored for 5 min with reads every 30 s. The GST activity was expressed in µmol of reaction product, GS-DNB conjugate (ɛ  =  9.6 mM^−1^ cm^−1^) per minute and gram FW. 

### 2.7. DPPH Radical Scavenging Assay

DPPH (2,2-diphenyl-1-picrylhydrazyl) free radical was used to evaluate the radical scavenging activity of *S. vulgaris* shoots [35]. The change in absorbance at 517 nm was measured just after the extract mixing with DPPH reagent, as well as after 30 min. Radical scavenging activity was calculated as percentage of reduced DPPH^•^ by a unit of extract.

### 2.8. HMs Detection in Tissue using Scanning Electron Microscopy (SEM)

The detection of HMs in plant tissue was performed on cross-sectioned surfaces of chosen leaves treated with 1× HMs and control ones. The taken leaf fragments were promptly mounted on a cryo-holder by using a cryo-embedding compound (OCT, Tissue-Tec, Sakura, Alphen aan den Rijn, The Netherlands) and colloidal graphite. Mounted material was submerged in liquid nitrogen and immediately put into the cryo-preparation chamber (Auriga 60, Zeiss). Platinum was used to sputter samples before observation in the cryo-SEM chamber at −140 °C and 2 kV of acceleration voltage, using SE2 and InLens detectors. The composition and localization of elements in vascular bundles and epidermal cells was performed at 20 kV of acceleration voltage with the use of an Oxford detector. Metals contents are presented in percentage by weight (wt. %).

### 2.9. Statistics

The whole experimental scheme was repeated three times. The measurement of growth parameters was conducted on 30 microcuttings per treatment within one replicate (in total, the 90 measurements were performed within one treatment). All physiological parameters were determined in three biological replications for particular treatments, each with 100 mg of appropriate shoot samples. The microscopic analyses were done on six leaf blade fragments of each ecotype taken from randomly chosen shoots cultivated on the control and on HM-enriched media.

Statistical analyses were performed using Statistica ver. 13.3 (TIBCO Software Inc., Palo Alto, CA, USA). Regarding biometric parameters, one-way ANOVA was applied separately for each ecotype as well as passage in the case of the micropropagation coefficient. Other data were evaluated by two-way analysis of variance (ANOVA) with ecotype and HM treatment as factors. The means were compared using Fisher’s test at *p* < 0.05. Such analysis was applied since preliminary calculations revealed that most physiological parameters depended on all tested factors, i.e., ecotype, treatment, and their combination. The exceptions were total secondary metabolites, flavonoids, and carotenoid contents as well as POD_8.8_, SOD, and APX activity, which depended only on both factor combination, and additionally ecotype in the case of total metabolites and flavonoids concentration and APX activity or treatment in the case of carotenoids. 

Moreover, principal component analysis (PCA) was used to analyze interrelationships among antioxidant parameters and cluster them into homogenous groups representing the most influential factors that differentiated responses between the tested ecotypes. 

## 3. Results

### 3.1. Culture Growth in the Presence of HMs

After four weeks of cultivation, MC values calculated for most NM cultures were comparable and ranged from 2.1 to 2.4, and only shoot cultured on medium with the highest HM dose (5×) showed a significant diminishing of propagation rate (MC = 1.1) (Figure 1A). The differences between treatments deepened after eight weeks, when a gradual MC decrease was observed with increasing metal concentration in the medium, and the lowest value of 1.1 was found for 2.5× and 5× HM-treated cultures. What is more, the highest concentration of applied HMs caused the death of over 75% of tested NM shoots. 

On the contrary, CAL shoot multiplication increased with the cultivation time on HM-enriched media, and MC values calculated after eight weeks for shoots growing in the presence of 2.5× and 5× HMs were almost two-times higher than after four weeks (Figure 1B). Interestingly, the multiplication of 1× HM-treated CAL culture remained at the constant high level of 4.2–4.3, independent of cultivation time, and finally reached a higher value in comparison to untreated culture (MC = 3.6). In turn, the differences in multiplication rate of SER shoots between control and cultures treated with 1× HMs were statistically insignificant in both evaluation periods, and amounted to MC = 2.0–2.4 after four weeks of treatment, and MC = 1.8–1.9 after eight weeks (Figure 1C). The doses of applied metals higher than 1× HMs resulted in severe growth disturbances of SER culture, that manifested in a significant decrease in multiplication efficiency on 2.5× HMs medium or even culture death on 5× HMs at the end of the experiment. Consequently, the biometric evaluation for this last treatment was omitted. 

Apart from the average number of newly formed shoots from one explant, the applied concentrations of HMs also influenced shoot length (Table 1). Regardless of the treatment, the values of this parameter were diversified due to the continual regeneration of shorter shoots, and therefore both minimum, average, and maximum, as well as median values are presented in Table 1. It was found that the increasing content of HMs in the medium reduced the length of NM shoots, however the mean values of this parameter in the control and 1× HMs-treated shoots were similar and varied from 22 to 28 mm. In CAL cultures, control shoots reached significantly lower, about 6 mm length, than shoots treated with 1× HMs, while shoots maintained on 2.5× HMs and 5× HMs media were comparable to each other but considerably shorter than untreated ones. Additionally, the spontaneous rhizogenesis was observed in CAL culture treated with 1× and 2.5× HMs. In other treatments as well as ecotypes, the adventitious root development was not found either on control or HM-enriched medium. The most pronounced changes in culture growth were noticed in SER shoots cultured in the presence of 2.5× HMs, for which the average length did not exceed 5 mm, whereas control shoots of this ecotype were almost ten times longer. 

The differences in shoot growth were also reflected in fresh and dry biomass (Table 1). In the NM ecotype the value of these parameters gradually decreased with the increasing HM content in the medium, from over 1200 to almost 350 mg for fresh weight and from 120 to 21 mg for the dry one. In the CAL ecotype, the highest biomass accretion was ascertained in 1× HMs-treated culture, then in the control one, while the lowest was in the 5× HMs-treated shoots. Despite this, the dry matter content of shoots cultured in the presence of 2.5× and 5× HMs was similar to each other and amounted, on average, to about 60 mg. In SER specimens, the highest values of fresh and dry matter content were noticed in the control culture, while the application of HMs reduced them significantly, independent of the treatment, up to an average of 60% of the control. 

### 3.2. Anatomy of HM-Treated Leaves

Taking into account the death of SER specimens on medium enriched with 5× HMs, this treatment was omitted for analyses regarding anatomical, ultrastructural, and biochemical features, and only 1× and 2.5× HMs-treated shoots were taken under consideration. 

Microscopic observations of leaf blade anatomy revealed the similarity in cell shape and size between untreated and 1× HMs-treated specimens of all tested ecotypes over the entire cross-section (Figure 2). Nevertheless, a more compact arrangement of cells on the adaxial than on the abaxial leaf side could indicate a distinguishing in mesophyll tissue. 

Regardless of ecotype, the typically differentiated parenchyma on the palisade and spongy layer was noticed in specimens growing on 2.5× HMs (Figure 2). However, in both NM and SER leaves a looser arrangement of cells with a slightly changed shape was observed. Additionally, the thickness of leaves varied between ecotypes and treatments. In NM, the thinnest leaves, of approximately 140 μm thick, were found in culture growing on medium containing 1× HMs, while in 2.5× HMs-treated and untreated ones it reached about 180 μm. In turn, in metallicolous cultures the leaves thickened with increasing HM content, from 130 μm in control, through 150 μm in 1× HMs-treated, to 200 μm in 2.5× HMs-treated CAL specimens, and from 215 μm, through 240 μm, to even 300 μm in SER specimens on control, 1×, and 2.5× HMs-enriched medium. Furthermore, numerous spherical structures probably of lipid origin, concentrated mainly near the vascular bundles, were noticed in CAL specimens cultivated on 1× and 2.5× HMs media (Figure 2). 

### 3.3. Photosynthetic Pigments’ Accumulation and Chloroplast Ultrastructure 

The toxicity of HMs to cultured explants was manifested in the gradual decrease in photosynthetic pigments’ content in NM and SER shoots, in which both chlorophyll *a* (chl *a*) and *b* (chl *b*) concentrations were lower than in the respective controls (Table 2). 

On the contrary, in CAL shoots the level of chl *a* remained stable, independent of the treatment, and ranged from 0.13 to 0.18 mg g^−1^ FW, however, a statistically insignificant increase was noticed in 1× HMs (Table 2). In turn, the chl *b* accumulation in CAL shoots declined, but it was comparable between the control and 1× HMs-treated cultures as well as the latter and 2.5× HMs one. Furthermore, the chl *a/b* ratio in NM culture significantly decreased in shoots from the 1× HMs medium and increased in shoots from the 2.5× HMs, while in SER shoots it was similar in both HM treatments and simultaneously lower than in the control one. In CAL shoots growing in the presence of HMs, calculated values of this parameter were almost two times higher than in the respective untreated shoots and differed significantly from each other, from 3.3 to 4.1 for 2.5× and 1× HMs-treated cultures, respectively. Regardless of applied doses of Zn, Pb, and Cd ions, the ratio of total chlorophyll to carotenoids (chl *a*+*b*/car) decreased considerably in NM and SER shoots, from HMs-enriched medium as compared to respective controls, while in CAL shoots it remained unchanged in all treatments (Table 2).

Cell ultrastructure analysis revealed that, independent of ecotype, chloroplasts in control leaves had regular structure, a few small plastoglobules, and huge starch grains (Figure 3A–C). 

On the contrary, chloroplasts from leaves of shoots cultured in the presence of different HMs doses varied significantly (Figure 3D–N). In 1× HMs-treated NM culture, chloroplasts that changed in shape, with a looser arrangement of thylakoid and numerous plastoglobules, were reported (Figure 3D), whereas in the 2.5× HMs-treated NM culture the disturbances were the most numerous and manifested additionally by chloroplast dilatation and the appearance of more abundant plastoglobules (Figure 3E). Generally, CAL chloroplasts showed no significant changes, providing the negative effects of metals on their ultrastructure. Only a slight reduction in the size of chloroplasts and larger plastoglobules in comparison to control ones was observed (Figure 3F–K). Additionally, in 1× HMs-treated leaves, among chloroplasts with starch grains (Figure 3F), those without them appeared, and then their granum thylakoids were relatively poorly developed (Figure 3G), while in 2.5× HMs-treated leaves starch grains were not found at all, but few chloroplasts with amorphous shape and looser arrangement occurred (Figure 3H). Unlike other ecotypes, chloroplasts of CAL leaves under both HM concentrations formed stroma-filled tubes, called stromules, that ran most often towards other chloroplasts, mitochondria, or the cell nucleus (Figure 3H–K). In turn, chloroplasts of HM-treated SER leaves exhibited an altered arrangement, swollen stroma, and dilated thylakoids (Figure 3L–N). These deformations were particularly visible under 2.5× HMs stress (Figure 3N), while in the presence of a lower HM dose the chloroplasts with a regular structure comparable to control were noticed more often. 

### 3.4. Lipid Peroxidation under Metallic Stress

Heavy metal harmfulness for cultured ecotypes was also evaluated on the basis of malondialdehyde (MDA) content, which is a commonly known marker of lipid peroxidation reflecting oxidative stress level. The MDA concentration increased significantly in HM-treated NM and SER shoots in comparison to respective controls, although to a lesser extent in SER culture on medium enriched with 1× HMs (Table 2). Interestingly, the oxidation status of CAL shoots remained unchanged after the application of 1× HMs, or even significantly decreased in shoots treated with 2.5× HMs compared to the control one. 

### 3.5. Non-Enzymatic Antioxidants in Silene Shoots Cultured on HM-Containing Media

The level of low molecular weight antioxidants, such as carotenoids and proline, depended on ecotype and HM-treatment (Figure 4A,B). Carotenoids content in NM shoots declined almost to 75% and 35% of the control in 1× and 2.5× HMs treatment, respectively (Figure 4A). In CAL shoots the concentration of carotenoids did not differ between treatments and amounted to about 0.06 mg g^−1^ FW, while in SER cultures growing on 1× HMs containing medium a significant drop, by 50% in comparison to other SER treatments, was observed. In turn, proline accumulation was the same in both NM HM-treated shoots, but simultaneously about two-times lower than in the control one, in which its content reached about 28 mg 100 g^−1^ FW (Figure 4B). In another way the concentrations of HM used in the present experiment influenced the proline content in shoots of metallicolous ecotypes. In CAL shoots a significant increase of about 50% of the control was noted only in shoots cultured on 1× HMs-enriched medium, whereas in 2.5× HMs-treated and untreated ones the proline level was comparable and amounted to about 8 mg 100 g^−1^ FW. In SER specimens, the amount of proline was the highest in control shoots (37 mg 100 g^−1^ FW), even in comparison to other ecotypes and treatments, but its accumulation was significantly reduced to 10 and 13 mg 100 g^−1^ FW in 1× and 2.5× HMs-treated shoots, respectively (Figure 4B). 

Results concerning compounds of phenolic origin showed that various HM doses also affected their metabolism in a particular ecotype (Figure 4C–G). Most of these examined parameters depended on all tested factors, i.e., ecotype, treatment, and their combination. However, the accumulation of total secondary metabolites as well as flavonoids was also determined only by ecotype origin. The content of polyphenols in NM and SER shoots gradually increased with the increasing concentration of Zn, Pb, and Cd ions in the propagation medium, and reached higher values than in control ones. Nevertheless, these polyphenols’ accumulation was independent of applied doses of HMs and a lack of statistical differences between both treatments was found. Similarly, in the CAL ecotype both HM doses considerably elevated the polyphenols content, however a significantly higher concentration of 8.55 mg g^−1^ FW was ascertained under 1× than 2.5× HMs stress (Figure 4C). The HM application significantly diminished the phenylpropanoids content in NM shoots, whose amount in SER specimens remained independent of metal presence in the medium and varied from 170 to 195 mg g^−1^ FW (Figure 4D). In the case of the CAL culture, an enhanced accumulation, up to 65% of the control, was once again noted in 1× HMs-treated shoots, while phenylpropanoids content in the 2.5× HMs treatment did not differ from the control and reached, on average, 140 mg g^−1^ FW. Taking into account the flavonols accumulation, it was noticed that their amount in NM shoots cultured on different media changed in an analogous way to phenylpropanoids content (Figure 4E). In CAL shoots the flavonols amount achieved the same level under both applied HMs doses, which was higher than in the control one, while in SER shoots a significant drop in accumulation was ascertained (Figure 4E). The amount of anthocyanins was only reduced in 2.5× HMs-treated NM shoots in comparison to control and 1× HMs-treated ones (Figure 4F). In turn, its concentration in CAL shoots was not altered under both HM treatments, and was even significantly higher—about 20–25 mg 100 g^−1^ FW—than in the untreated control, which contained 36 mg of anthocyanins per 100 g of FW, whereas in SER cultures it decreased significantly with the increasing content of toxic ions in the medium. Moreover, the sum of secondary metabolites with a double bond in their chemical structure changed in NM shoots in the following order: control > 1× HMs > 2.5× HMs (Figure 4G). In turn, CAL shoots showed a significant increase in total secondary metabolites under 1× HMs treatment, while untreated and 2.5× HMs-treated shoots exhibited lower but similar to each other accumulation in the range of 470–520 mg 100 g^−1^ FW. Interestingly, the pattern of these compounds’ production in SER shoots showed an opposite trend to the CAL ecotype, and a higher accumulation was noticed in control and 2.5× HMs-treated shoots than in 1× HMs-treated ones. Finally, radical scavenging activity in NM and CAL cultures seemed to be correlated with the anthocyanin accumulation, which showed a similar pattern of changes, whereas in the SER ecotype such a relationship was not observed (Figure 4H). 

### 3.6. Antioxidant Enzyme Responses of Silene Shoots to HMs 

To investigate the antioxidant system efficiency under HMs stress, the activity of chosen enzymes was determined (Figure 5A–G). The measurements of superoxide dismutase (SOD) activity revealed its significant decrease by 35% in NM shoots on 1× HMs-containing medium in relation to shoots untreated with HMs, as well as treated with 2.5× HMs, in which this enzyme activity was comparable and amounted to 32 U g^−1^ FW (Figure 5A). In CAL cultures, the application of a higher HM concentration influenced the enhancement of SOD activity in comparison to control shoots, however the difference between particular HMs treatment was statistically insignificant. In the case of SER cultures, SOD activity remained unchanged, irrespective of HM presence in the medium. 

The catalase (CAT) activity in NM and CAL shoots was comparable in individual ecotypes between HM-treatments, and reached the level of 1.6 and 3.3 μmol min^−1^ g^−1^, respectively (Figure 5B). However, as compared to the respective control, CAT activity decreased in NM and increased in CAL shoots under HM stress. In SER shoots, CAT activity declined to 62% and 44% of the control in 1× HMs- and 2.5× HMs-treated shoots, respectively. 

The HM application contributed to significant changes in various peroxidases activity in an ecotype-dependent manner (Figure 5C–F). In NM specimens, guaiacol-type peroxidase (GOPX) as well as peroxidases measured in pH 6.8 (POD_6.8_) and pH 8.8 (POD_8.8_) exhibited the lowest activity in 1× HMs-treated shoots (Figure 5C–E). In turn, their highest values were ascertained in control shoots in the case of GOPX and in 2.5× HMs-treated shoots in the case of POD_6.8_, whereas the activity of POD_8.8_ was the same in shoots on control and 2.5× HMs-enriched medium. On the contrary, the ascorbate peroxidase (APX) showed the opposite trend and its activity reached the highest value of 188 μmol min^−1^ g^−1^ in shoots growing on medium supplemented with 1× HMs, while the lowest was in the control one (Figure 5F). Considering CAL specimens, the activity of GOPX, POD_6.8_, and POD_8.8_ was the highest in shoots cultured in the presence of 1× HMs, where it increased by three- and two-times over control for GOPX and POD, respectively (Figure 5C–E). Interestingly, APX level in CAL shoots remained unchanged even if HMs were applied, although its activity was relatively high in comparison with other ecotypes (Figure 5F). In SER culture, GOPX activity in shoots multiplicated on 1× HMs-containing medium was enhanced over 2.5-times compared to shoots from medium supplemented with 2.5× HMs, in which the activity of this enzyme did not differ from the control and amounted to 1.3 μmol min^−1^ g^−1^ (Figure 5C–E). Regardless of the pH measurement, POD activity in SER 1× HMs-treated shoots was as high as in the control, whereas in 2.5× HMs-treated ones POD_6.8_ declined significantly in comparison to control culture or did not change in the case of POD_8.8_. A similar tendency was observed for APX, albeit its activity in the presence of the highest HMs doses was the opposite to POD, and thus it rose almost two times relative to the control and 1× HMs-treated shoots (Figure 5F). 

Surprisingly, the glutathione-*S*-transferase (GST) activity was detected primarily in the NM ecotype (Figure 5G). In this case, the enzyme activity increased 1.6-times in 1× HMs-treated NM shoots as compared to untreated ones, in which it reached the level of 433 μmol min^−1^ g^−1^, while in 2.5× HMs-treatment it declined to just 35% of the control. In CAL untreated shoots GST activity was similar to NM untreated shoots. Despite this, a complete lack of activity or a significant reduction by 56% in comparison to the control was observed in 1× HMs- and 2.5× HMs-treated CAL shoots, respectively. In turn, SER cultures growing on medium without HMs as well as on medium enriched with 1× HMs were characterized by GST activity below the detection threshold. Nevertheless, the addition of 2.5× HMs resulted in an increase in activity to the level comparable with CAL shoots treated with the same HMs dose. 

### 3.7. Major Players in Antioxidant Machinery of Particular Ecotypes

A principal component analysis (PCA) was conducted separately for NM, CAL, and SER cultures in order to find the specific and unspecific chemical compounds involved in antioxidant response under particular HM doses. Figure 6 displays the plots of tested variables and treatments in terms of PC 1 and PC 2. 

For the NM ecotype, the first and the second component explained 54% and 36% of the total variability, respectively (Figure 6A). Most of the non-enzymatic antioxidants were strongly positively correlated with PC 1. The exception was the polyphenols content, which exhibited a negative, but very strong relationship with this factor. Moreover, among phenolic compounds, anthocyanins were proven to have the highest activity to scavenge the DPPH radical. The PC 1 was also determined by CAT and GST, whereas other antioxidant enzymes were strongly positively correlated with PC 2, although in the case of APX the direction was negative. Additionally, APX was negatively correlated with GOPX and POD_8.8_, which in turn exhibited a positive relationship at a strong level with each other, as well as a weak relationship with SOD and POD_6.8_. Taking into account particular treatment, it could be noted that most of the tested variables composed a close association with control culture. Despite this, the medium supplementation with 1× HMs strongly affected the activity of APX and GST, while the addition of 2.5× HMs influenced POD_6.8_ activity in NM shoots (Figure 6A). The unspecific reaction of the NM ecotype to HMs relied on polyphenols, whose the content changed in similar way in both treatments. 

Unlike NM specimens, in the CAL ecotype the biggest association, comprising the majority of antioxidants, was closely positively related with HMs-treated shoots, particularly in the 1× HMs dose (Figure 6B). For this treatment, the accumulation of proline, polyphenols, phenylpropanoids, as well as the enhanced activity of enzymes representing the peroxidase family (with the exception of APX) seemed to be the specific reaction to 1× HMs. In most cases, this group of variables was strongly negatively correlated with PC 1, which determined about 55% of the total variability, while their relationship with PC 2 was differentiated and rather low or moderate. On the contrary, PC 1 was positively correlated with GST, for which activity was associated with control shoots, as well as with APX; however, this variable could be recognized as weak factor loading for the CAL ecotype and thus it was omitted in further consideration. Among the tested variables the strongest positive correlation with PC 2 was determined for SOD, which activity was enhanced in shoots growing in the presence of 2.5× HMs. Meanwhile, the accumulation of anthocyanins and flavonols together with increased CAT activity would be associated with an unspecific response of CAL cultures to both applied HMs doses. Regardless of the treatment, in the CAL ecotype a positive relationship was determined between GOPX, POD_6.8_, and POD_8.8._ In turn, all compounds of phenolic origin were positively correlated with the DPPH radical, however the strongest relationship was determined for anthocyanins.

The principal component analysis for SER ecotype showed that the first two factors explained over 80% of the total variability (PC 1: 46%; PC 2: 35%) (Figure 6C). PC 1 was strongly negatively correlated with the polyphenols’ content, the radical scavenging activity, and chosen enzymes, such as APX, GST, while a positive correlation of PC 1 was noticed for anthocyanins accumulation as well as CAT and POD_6.8_ activity. In turn, the second factor was negatively correlated mainly with the content of flavonols, carothenoids, and proline, whereas GOPX exhibited a strong but opposite relationship with PC 2. Considering particular treatment, SER shoots growing on the medium without HMs composed a close association with the content of proline, anthocyanins, and flavonols, and simultaneously with the activity of CAT. The application of 1× HMs strongly affected GOPX activity, while the addition of a higher HM dose influenced mostly the APX and GST activity. The unspecific response to both HM treatments was based on polyphenols, which were additionally strongly correlated with the radical scavenging activity. Interestingly, such a relationship was not observed in other ecotypes. What is more, among the analyzed parameters in SER specimens the content of total secondary metabolites and phenylpropanoids as well as SOD activity had the lowest contribution to both principal component constructions.

### 3.8. Cryo-SEM-EDX Analysis

Since cultures of all ecotypes treated with 1× HMs remained alive and developed to the end of the experiment, the detection of HMs in plant tissue was performed only for this metal concentration. In general, the metal uptake proceeded similarly in all tested specimens (Figure 7). Regardless of ecotype, leaf surface and main vascular bundles comprised 0.4%–0.5% wt. and 0.2%–0.3% wt. of Pb ions, respectively. This metal was also observed in NM and SER trichomes, where it reached a similar value of 0.4% wt in both ecotypes. In turn, Cd ions were not determined, while the Zn low level of 0.1% wt. was detected only in CAL vascular bundles and SER epidermis. 

## 4. Discussion

### 4.1. Morphogenetic Response of HM-Treated Cultures is Diversified

An interesting feature of *Silene vulgaris* is its occurrence in various, often very harsh environments, since apart from ecotypes tolerant to HMs, there are others resistant to salinity and drought [9,36,37]. Such exceptional adaptability of this species to abiotic stress factors makes *S. vulgaris* suitable for comparative analysis of the mechanisms underlying tolerance to severe conditions, as well as to elucidate the complexity of speciation phenomenon. In this study, *S. vulgaris* specimens representing non-metallicolous and two metallicolous ecotypes were used to investigate the impact of Zn, Pb, and Cd ions on growth, development, and antioxidant responses. As an experimental model, shoot culture was applied in order to standardize explant exposure to the tested factor due to the elimination of roots, which constitute a serious barrier in HM uptake and translocation to aboveground parts [38]. This assumption was fulfilled in our experiment, in which rhizogenesis was not noticed, and the accumulation of HMs followed the same pattern in all ecotypes. The exception was CAL cultures, which developed several adventitious roots in the presence of both applied HM doses, however the spontaneous rhizogenesis had no effect on metal uptake and could be regarded as one of the specific defense reactions to metal treatment. Surprisingly, although Pb was considered the least mobile among the tested elements, its ions were collected in the highest amount. Probably the well-known antagonistic relationship of Pb with Cd and Zn ions contributed to the reduced accumulation of the latter, which reached a level below the detection threshold by microscopic methods. Despite this, the reaction of particular cultures varied widely. 

The multiplication rate of NM shoots declined gradually with time and with increasing HM content in the medium, resulting in disturbances in culture growth and development, even at the lowest applied concentration of metals. In turn, SER cultures showed disturbances after the application of 2.5× HMs, but they appeared with the lapse of time, whereas the treatment with 5× HMs was proven to be lethal for this metal-tolerant ecotype at the beginning of the experiment. Adverse effects of metallic stress on plant morphogenesis have been demonstrated not only in the presented research, but also in other species belonging to various taxa. For example, the research of Malar et al. [39] on sensitive *Eichhornia crassipes* specimens revealed the negative impact of Pb ions at the concentration of 100 mg L^−1^ on shoot and root growth, while Glińska et al. [40] observed *Triticum aestivum* growth inhibition when 300 mg L^−1^ of Zn was used. Therefore, our results confirmed the sensibility of the NM ecotype, whereas the reaction of the SER one was quite interesting since SER specimens were found not to be fully adapted to the applied HMs. In contrast to them, the growth and multiplication rate of CAL specimens were stimulated under 1× HMs treatment. Such positive effects on CAL culture caused just by these HMs that are characteristic for calamine substrate, and simultaneously in the dose corresponding to their amounts available for plants on waste heap from tested ecotype originated, may implicate the increase requirement of CAL specimens for Zn, Pb, and Cd, i.e., those trace elements to which this ecotype has become adapted. This hypothesis was announced for the first time by Antonovics et al. [41], and confirmed in later study on other metal-tolerant taxon from the *Caryophyllaceae* family—*Dianthus carthusianorum* [6]. To some extent it also explains the recently postulated phenomenon of a dose-dependent response to HMs, called hormesis. According to this concept, metallic elements with unknown physiological function may have a beneficial impact on organisms when they are applied in low concentration, which has been observed not only for the tested CAL ecotype of *S. vulgaris*, but also for species such as *Brassica napus* treated with Cd ions [5] or *Pisum sativum* under Pb exposition [4]. Therefore, the applied HMs doses, which induced various disturbances in other ecotypes, did not exceed the toxic thresholds for the CAL one and may be considered as trace elements for proper growth and development.

To sum up, the results on macroscopic observations of culture growth may indicate the lack of cross-tolerance in metallicolous *S. vulgaris* specimens, which positively reacted only to those metals to which they have been adapted. However, to better explore this preliminary statement, experiments on these contrasting ecotypes treated with ions characteristic for serpentine soils (Ni, Co, Cr) are worth carrying out in the near future. What is more, our findings suggest that the development of tolerance in both metallicolous ecotypes may have resulted from a distant evolutionary pattern, although the selection factor, i.e., metals, seems to constitute the same group of stressor. In this context, a deeper insight into the ecotype-specific pathways of minimizing negative HM impacts is an extremely important issue to learn more about tolerance mechanisms. 

### 4.2. Interrelation of Structural and Physiological Features Responsible for Morphogenetic Reactions 

A key factor in the appropriate understanding of plant response to HM toxicity is the assessment of the interrelated features at different levels. Among many studies on physiological response to HMs, little information exists about leaf anatomical traits under stress conditions, which are linked to water status and CO_2_ assimilation rate, and might indirectly lead to changes in the photosynthesis [42,43]. Our research revealed that leaves of *S. vulgaris* exhibited great plasticity with respect to lamina thickness and mesophyll differentiation, depending on the treatment. Leaves of shoots treated with 1× HMs differed from the anatomical structure of leaves cultured in the presence of 2.5× HMs as well as those taken from natural habitats [22]. The mentioned traits refer to uniform multilayered mesophyll with closely packed cells and stomata distributed across both leaf surfaces. Such an increase in the number of photosynthesizing cells per unit of leaf surface and facilitated CO_2_ diffusion across the mesophyll due to amphistomatic leaves may suggest an attempt to adapt plant life processes to stress conditions and contribute to enhanced rates of transpiration and photosynthesis [44]. Despite this, NM and SER *Silene* cultures showed typical symptoms of metal toxicity, reflected in the reduced amount of pigments and changes in the chl *a*/*b* ratio. A similar relationship between pigment accumulation and resistance level was ascertained for numerous species, since a harmful impact of various stress factors is frequently manifested in a decrease in photosynthetic pigment content [37,45,46]. The decline in pigments in NM and SER ecotypes was accompanied by the disappearance of chloroplast ultrastructure, whose thylakoid arrangement became very loose. Instead, many plastoglobules, thylakoid-associated lipid droplets, occurred independently of metal dose, indicating the destruction of the inner membrane system in chloroplast, as previously observed, among others, in *Brassica napus* [47] or *Triarrhena sacchariflora* [48] treated with Zn or Pb and Cd ions, respectively. However, the presence of numerous plastoglobules in chloroplasts of SER leaves from the 1× HMs-treated culture, which did not show so many significant changes in the internal organization, may be related to their function in the stabilization of photosynthetic apparatus and suggest an attempt to remodel chloroplast membranes under stress conditions [49,50]. The disturbed accumulation of pigments could result also from the inhibition of their biosynthesis pathways due to the high redox potential of many HMs, which, for example, decline the activity of the key enzyme, protochlorophyllide reductase, involved in the reduction of protochlorophyll to chlorophyll [46,51]. Additionally, as observed, among others, for many water plants growing in the presence of metallic ions [52], the substitution of Mg^2+^ in the chlorophyll molecules by Zn^2+^, Pb^2+^, or Cd^2+^ may also occur regardless of the ecotype, and thus the pigment content in HM-treated shoots decreased significantly in comparison to control ones. The exception was CAL shoots growing on medium enriched with 1× HMs, in which the enhanced accumulation of chlorophyll *a* was noticed, while in the 2.5× HMs treatment this pigment content was comparable to control culture. It is also worth emphasizing that a negative impact of applied HM doses on chloroplast ultrastructure in this ecotype was not frequently observed and concerned mostly plants cultivated on medium supplemented with higher HM concentration. What is more, chloroplast with huge starch grains occurred in 1× HMs-treated leaves. Through combining chloroplast ultrastructure with anatomic traits and pigment content it can be inferred that Zn, Pb, and Cd, when applied in concentrations reflected in their content in the natural environment, stimulate photosynthesis and other metabolic processes in the CAL ecotype, since increased production of secondary metabolites and enhanced activity of antioxidant enzymes also occurred. Such efficient metabolism requires a large amount of energy. Thus, it is highly probable that the huge electron-dense lipid droplets, visible even under light microscope, were accumulated as a reservoir of fatty acids for beta oxidation, during which they are broken down to produce energy. These results may implicate the existence of alternative pathways of energy acquisition in the CAL ecotype that have not been noticed yet for any other metal-tolerant plants. However, this needs to be taken under deeper examination in the future. 

### 4.3. Stromules as Unique Feature of CAL Chloroplasts

An interesting feature of CAL chloroplasts that distinguishes them from chloroplasts of other ecotypes is the formation of stromules under the influence of metallic stress. Although these dynamic protrusions of the plastid envelope membrane can be formed under both unstressed and stressed chloroplasts [53,54], little is known about the mechanisms of their formation. Similarly, their roles in plant cells remain still quite enigmatic, despite numerous studies being conducted. Stromules were found to be involved in facilitating the transport of macromolecules or signals such as proteins, salicylic acids, or H_2_O_2_, which must pass from one location to another [40,55,56]. In this context, stromules may reduce the diffusion distance between cell compartments that exchange substrates and products for another metabolic pathway. According to Schattat and Klösgen [57], stromules participate in carbohydrate metabolism. This hypothesis could be supported by our study showing the presence of stromules only in chloroplasts without starch grains, which were probably broken down to pyruvate. The latter undergoes transformation into malate, which, in turn, is transported via cytosol to mitochondrion, where it constitutes a substrate for the Krebs cycle. Therefore, in CAL specimens the improvement of cellular respiration by an alternative pathway by passing the glycolysis may provide additional energy to maintain the metabolism as high as possible, in spite of stressor presence. On the other hand, it cannot be excluded that stromules in CAL chloroplasts appearing in cells with a cytoplasm that was slightly more electron-translucent than in control ones and often possessed multilamellar microvesicles may suggest the first symptoms of leaf cell degradation. Taking into account that stress conditions induce ROS generation, for which internal changes in chloroplast were proven to participate in stromule formation [58], their frequency often increases in response to various abiotic stresses, such as drought, salt, and cold treatment [53,59]. In turn, Glińska et al. [40] noticed the protrusion of *Triticum aestivum* chloroplasts resulted from degradation processes under an excess amount of Zn. However, in this case, disturbances of cell ultrastructure were also manifested by membrane disintegration or mitochondrion alternations. On the contrary, in CAL specimens the observed changes underlay the senescence processes rather than resulting from the negative impact of HMs, because ultrastructural disorders occurred in adjacent cells, next to which cells without degenerative symptoms were found. Additionally, the marker of membrane injury (MDA) still remained unchanged in 1× HMs-treated specimens and even diminished in 2.5× HMs-treated ones. Since Gregersen et al. [60] summarized that during senescence partially degraded proteins of chloroplasts are removed from the plastids by the release of vesicles from stromules, their formation in CAL specimens could result from increased proteolytic activities. Thus, stromules of CAL chloroplasts would participate in increasing the surface area to promote macromolecular trafficking. However, more detailed findings on proteolysis will soon be presented in a separate article. 

### 4.4. Changes in Antioxidant Accumulations are Ecotype-Specific

Abiotic stresses have been demonstrated to accelerate reactive oxygen species (ROS) generation. To mitigate their harmful effects, plants possess an interrelated network of antioxidant mechanisms [61,62,63]. In the current study, *S. vulgaris* shoots of NM and SER ecotypes cultured in the presence of HMs experienced oxidative stress, reflected in the enhanced lipid peroxidation measured by MDA content. On the contrary, in the CAL ecotype MDA accumulation was similar between control and 1× HMs-treated shoots, or even decreased in 2.5× HMs-treated shoots, indicating ROS homeostasis and/or a higher tolerance than others to applied metals. 

The tested cultures modified ROS deactivation pathways in an ecotype-specific manner, although some antioxidant strategies were common to all specimens. To visualize these responses to particular HM doses, a summary of examined non-enzymatic and enzymatic antioxidants, that changed or not in NM, CAL, and SER ecotypes, is shown in Figure 8. 

Regardless of ecotype, polyphenol content increased in all shoots growing on HM-enriched media. The synthesis of these antioxidants is frequently strengthened under environmental stressors, and therefore they play an important role in the regulation of plant reaction to various abiotic and biotic stressors [63,64,65]. Accordingly, in the examined ecotypes polyphenols were found to be a universal group of compounds for *S. vulgaris* ecotypes. However, only in SER specimens polyphenols were correlated with free-radical scavenging activities, suggesting that different compounds from this heterogeneous group of secondary metabolites might be synthetized in particular ecotypes. 

Apart from enhanced accumulation of polyphenols, the content of other ROS non-enzymatic scavengers decreased in NM and SER shoots treated with HMs in comparison to untreated controls (Figure 8). This could imply their consumption for defense mechanisms accompanied by the inhibited activities of enzymes involved in phenolic biosynthesis, as well as incorporation into other metabolic pathways, as in the case of proline. The exception was phenylpropanoids in SER shoots, whose level remained constant (statistically insignificant) even after HM application. Similarly, their concentration in CAL 2.5× HMs-treated shoots did not differ remarkably compared to control culture, however a slight increase in their accumulation was ascertained.

On the contrary, the application of 1× HMs during CAL cultivation resulted in a significant accumulation of this group of phenolic compounds that are involved in lignification and/or suberification [66]. Taken together with the highly enhanced activity of guaiacol peroxidase (GOPX) in this treatment, which consumes H_2_O_2_ in order to generate phenoxy compounds used for production of cell wall components [10,67], the building of a thicker mechanical barrier preventing ion penetration by cell wall lignification could be attributed to the defense strategy in the CAL ecotype against metals at concentrations equivalent to their natural content (i.e., 1× HMs). The highly increased activity of GOPX was also observed in 1× HMs-treated SER shoots, however it was not accompanied by changes in phenylpropanoid accumulation. It is highly probable that in the SER ecotype GOPX is associated with detrimental stress only by consuming H_2_O_2_, while the lethal dose of 2.5× HMs leads to the activation of APX for H_2_O_2_ scavenging as well as to the synthesis of other secondary metabolites with double bonds in their chemical structure. They could be organic acids, which were found to play a crucial role in response of *S. vulgaris* to zinc or chromium, as both metal-binding compounds and ROS scavengers [36,68,69]. On the other hand, in CAL leaves metallic ions may be also detoxified by chelation with subgroups of flavonoids, such as flavonols and anthocyanins, whose accumulation increased remarkably after HM addition in a dose-independent manner. This statement seems to be particularly important, taking into account that metal–flavonoid complexes have been found to operate as more efficient antioxidants than the initial flavonoids [16]. Moreover, flavonoids are able to quench H_2_O_2_ freely diffusing from other compartments to vacuoles, when APX and CAT activity is suppressed [70]. In our study, APX activity in CAL shoots treated with HMs did not change with respect to untreated ones, indicating the insignificant role of this enzyme in ROS deactivation, whereas CAT activity increased similarly in both HM-treated CAL shoots, about 30% of the control. Since CAT has a very low affinity for H_2_O_2_ and requires two molecules of this substrate in the active site, it better controls the high level of H_2_O_2_ in the cells [71]. Therefore, our experiment may indicate a low H_2_O_2_ concentration in CAL leaves, due to their effective elimination or consumption for lignin biosynthesis, which remains at the level important only for regulatory mechanisms that engage stromules as a signaling transmitter. In turn, in NM and SER shoots treated with HMs, the gradual decline in CAT activity with the increasing metal content in the medium might be attributed to the inhibition of this protein synthesis or the engagement of APX in H_2_O_2_ management in NM specimens, since APX activity remarkably increased after HM application in this ecotype. APX overexpression was particularly noticed in 1× HMs-treated NM shoots, in which various peroxidases, as other antioxidant enzyme class responsible for H_2_O_2_ decomposition, were depleted. Therefore, APX may act as a specific ROS scavenger in the NM ecotype, which is assisted depending on metals dose by GST and POD_6.8_ activity. Nevertheless, such a defense mechanism based on a limited number of enzymatic and non-enzymatic antioxidants was proven to be insufficient for HM-stress protection resulting in morphological, anatomical, and physiological disturbances of NM cultures. 

In general, enzymes from the peroxidase family can be used to anticipate the response to the organism level since they are sensitive indicators of HM stress [62,72]. Particular peroxidases exhibit activity in various pH, depending on their cellular localization. GOPX are located in the cytosol, vacuole, and cell wall, while others are found in the cytosol and chloroplasts [73]. In the CAL culture the activity of GOPX, POD_6.8_, and POD_8.8_ changed significantly in an HM-dose dependent manner, i.e., it increased significantly in 1× HMs-treated shoots and decreased at higher metal concentrations (Figure 8). There is evidence that elevated levels of these scavenging enzymes provide antioxidative protection to combat the negative consequences of HM stress in the CAL ecotype. The ability of plants to effectively counteract the harmful effects of excess HM concentration is rather limited, and many studies showed the opposite tendency, in which decreased peroxidase activity was manifested in growth disturbances [74,75]. Such a relationship could also be confirmed in the SER ecotype which exhibited only a relative increase (statistically insignificant) in peroxidase activity in shoots cultivated on 1× HMs-enriched medium and their relative decrease in 2.5× HMs treatment in comparison to shoots from the control medium (Figure 8). Considering that other symptoms of HM toxicity in SER specimens resulted even in culture death as well as a reverse reaction of CAL ones, our findings give an impression of the high importance of peroxidases in metallicolous ecotypes’ response to metallic ions. In turn, SOD activity in SER and CAL specimens seems to be irrelevant in the reaction to HMs, although its overproduction is often correlated with enhanced tolerance to various environmental stressors [10,39,76]. This enzyme catalyzes the inactivation of O_2_^•−^ - the primary ROS formed in the cells [63]. Therefore, the observed lack of significant differences in SOD activity between treatments probably resulted from earlier dismutation of O_2_^•−^ to H_2_O_2_, which, in turn, was eliminated in subsequent antioxidant reactions. What is more, we demonstrated that the increased MDA level in HMs-treated NM and SER shoots was accompanied by the enhanced GST activity. Thus, it is likely that in these ecotypes the fatty acid hydroperoxides formed during lipid peroxidation were effectively conjugated to reduced glutathione molecules to be detoxified to hydroxy derivatives [77].

## 5. Conclusions

The present research demonstrated the integrated structural, biochemical, and morphological response of *S. vulgaris* ecotypes to HMs, which resulted from the activation of the antioxidant defense system. The analyzed pathways of ROS scavenging relied mostly on the non-enzymatic and enzymatic antioxidants specific to the respective ecotypes, and these unique antioxidant components seem to be particularly interesting in terms of plant acclimation and adaptation to metallic stress.

The highest ability to counteract oxidative stress, reflected in the improvement of physiological and morphological parameters, was ascertained in the CAL culture. Specimens of this ecotype showed two opposite strategies to cope with an excess amount of HMs. One of them functioned at a lower HM concentration, and apart from ROS deactivation, it was based on the prevention of HMs entry to the protoplast by cell wall lignification (the activation of peroxidases and phenylpropanoid synthesis involved in this process). The second strategy was independent of HM dose and played a role inside the protoplast, where metallic ions might be chelated with subgroups of flavonoids to form complexes that are even more effective in ROS deactivation. Therefore, the anthocyanins and flavonols together with CAT activity can act irrespectively of HM concentrations as crucial components of the CAL response to metal presence. Interestingly, SER specimens treated with 1× HMs reacted quite similarly to CAL specimens, and exhibited enhanced activity of enzymes from the peroxidase family. In turn, the application of a higher HM dose induced a reaction analogous to the NM culture response, and only the activity of APX and GST increased significantly in this treatment. This points to the low potential of these compounds for antioxidant protection under stress conditions, since both SER and NM shoots showed growth disturbances combined with ultrastructural alternations, as well as a decline in photosynthetic pigment content.

To sum up, our findings provide a better understanding of the operation of antioxidant machinery in metallicolous and non-metal tolerant specimens and place peroxidases among the best antioxidants. Undoubtedly, this issue is worth exploring in the near future in order to reveal both specific and conservative isoenzymes engaged in adaptative mechanism to HMs. Therefore, our study may give suggestions that facilitate obtaining plants adapted to environmental stressors through the manipulation of the antioxidant system and enables a further increase of plant productivity in harsh conditions. 

## Figures and Tables

**Figure 1 antioxidants-09-00102-f001:**
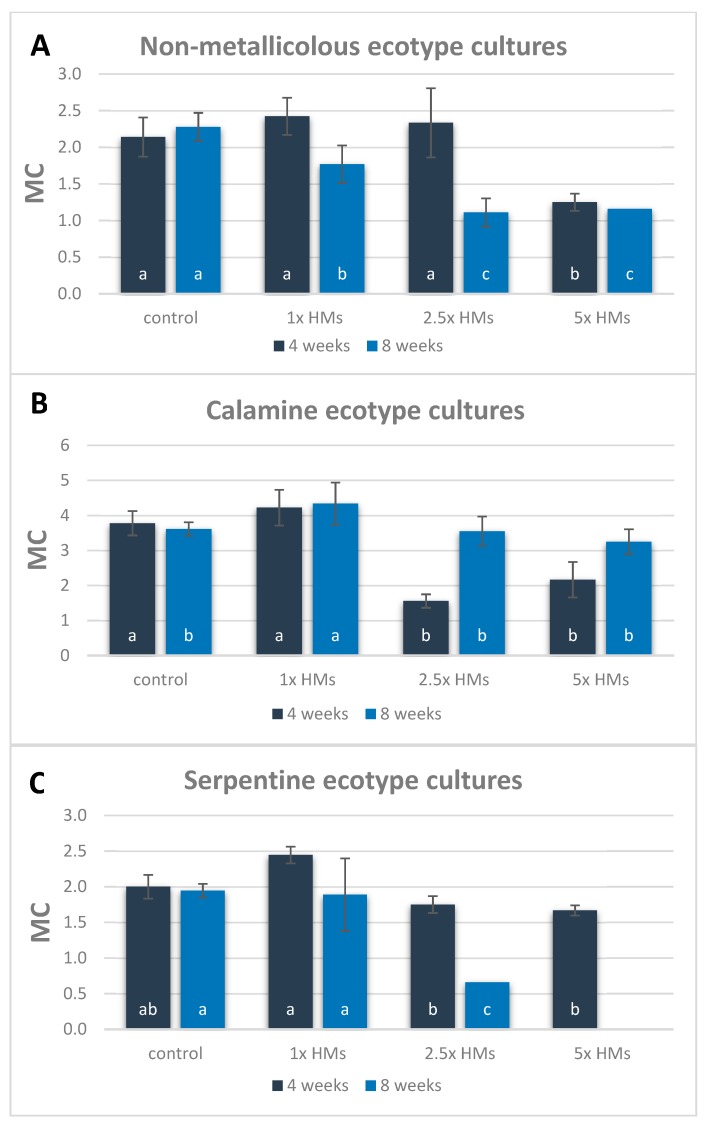
Multiplication coefficient (MC) for non-metallicolous (**A**), calamine (**B**) and serpentine (**C**) ecotypes of *Silene vulgaris* calculated separately for four and eight weeks of cultivation on media enriched with different heavy metal (HMs) concentrations. Values are means ± SD. Means indicated by the same letters (a–c) within each cultivation time point show statistically insignificant differences according to one-way ANOVA and post-hoc Fisher’s test at *p* < 0.05. MC represents the ratio of the newly regenerated shoots to initially used explants. Control means the medium without the addition of Zn, Pb, Cd ions; 1× HMs—medium enriched with 714.3 μM ZnSO_4_, 3.0 μM Pb(NO_3_)_2_, and 16.4 μM CdCl_2_; 2.5× and 5× HMs mean, respectively, 2.5- and 5.0-times higher Zn, Pb, Cd concentration than the first one (1× HMs).

**Figure 2 antioxidants-09-00102-f002:**
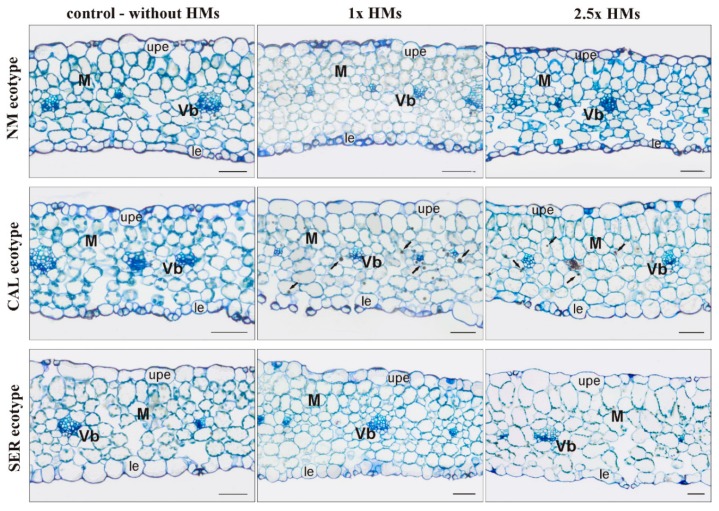
Cross sections of non-metallicolous (NM), calamine (CAL), and serpentine (SER) leaves taken from *Silene vulgaris* specimens cultivated on control medium without heavy metals (HMs) as well as on medium enriched with 714.3 μM ZnSO_4_, 3.0 μM Pb(NO_3_)_2_, and 16.4 μM CdCl_2_ (1× HMs) and 2.5-times higher (2.5× HMs) than the first one. Abbreviations: le—lower epidermis; M—mesophyll; upe—upper epidermis; Vb—vascular bundle; arrow—spherical structures of lipid origin. Bar = 50 μm.

**Figure 3 antioxidants-09-00102-f003:**
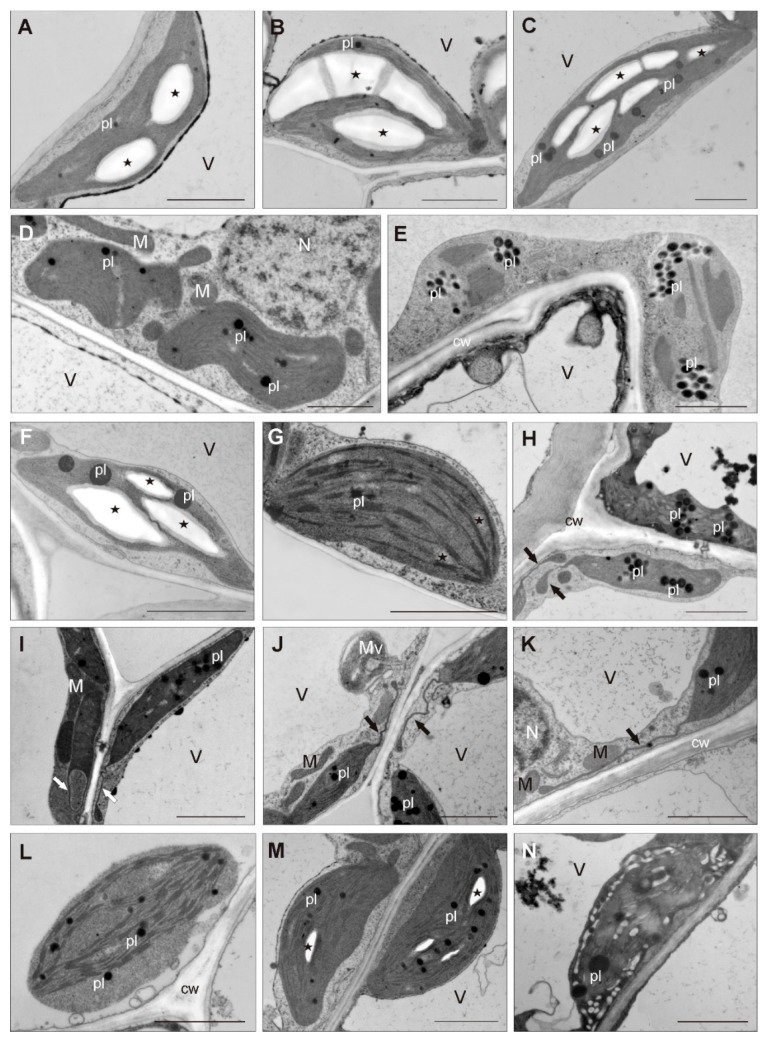
Chloroplasts of *Silene vulgaris* leaf mesophyll cells. **A–C**: Arrangement of chloroplasts in non-metallicolous (**A**), calamine (**B**), and serpentine (**C**) specimens growing on medium without heavy metals (HMs); **D–E**: chloroplasts of non-metallicolous plants treated with 1× (**D**) and 2.5× HMs (**E**) with a looser arrangement of thylakoids and numerous plastoglobules; **F–G**: properly developed chloroplasts with starch grains belonging to calamine specimens cultivated in the presence of 1× HMs; **H–I**: amorphous in shape chloroplasts of calamine 2.5× HMs-treated cells, close to which chloroplasts with stromules occurred; **J–K**: stromules formed by chloroplasts of calamine plants from medium containing 1× HMs; **L–N**: altered chloroplasts with swollen stroma and dilated thylakoids from serpentine plants growing in the presence of 1× (**L**–**M**) and 2.5× HMs (N). Abbreviations: cw—cell wall; N—nucleus; M—mitochondrion; pl—plastoglobule, V—vacuole; Mv—microvesicle; arrow—stromule; asterisk—starch grain. Bar = 2 μm.

**Figure 4 antioxidants-09-00102-f004:**
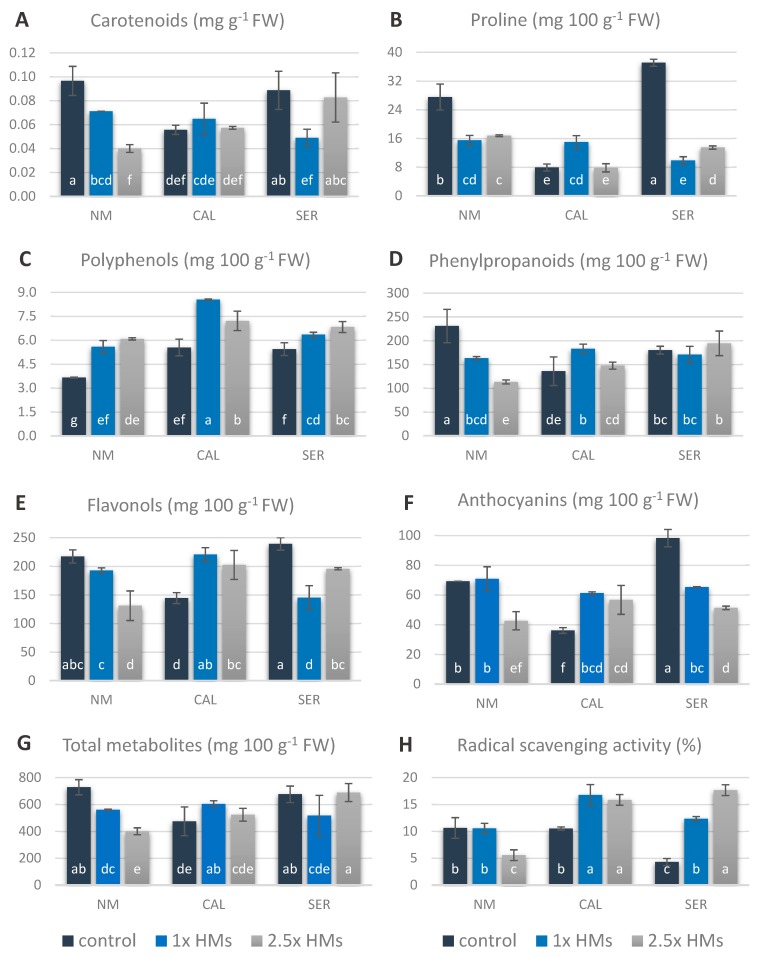
Non-enzymatic antioxidants in shoots of non-metallicolous (NM), calamine (CAL), and serpentine (SER) *Silene vulgaris* ecotypes treated with heavy metals (HMs). Values are mean ± SE. Means indicated by the same letters (a–f) point to statistically significant differences according to two-way ANOVA and post-hoc Fisher’s test at *p* < 0.05. Control means the medium without the addition of Zn, Pb, Cd ions; 1× HMs—medium enriched with 714.3 μM ZnSO_4_, 3.0 μM Pb(NO_3_)_2_, and 16.4 μM CdCl_2_; 2.5× HMs means 2.5-times higher Zn, Pb, Cd concentration than the first one.

**Figure 5 antioxidants-09-00102-f005:**
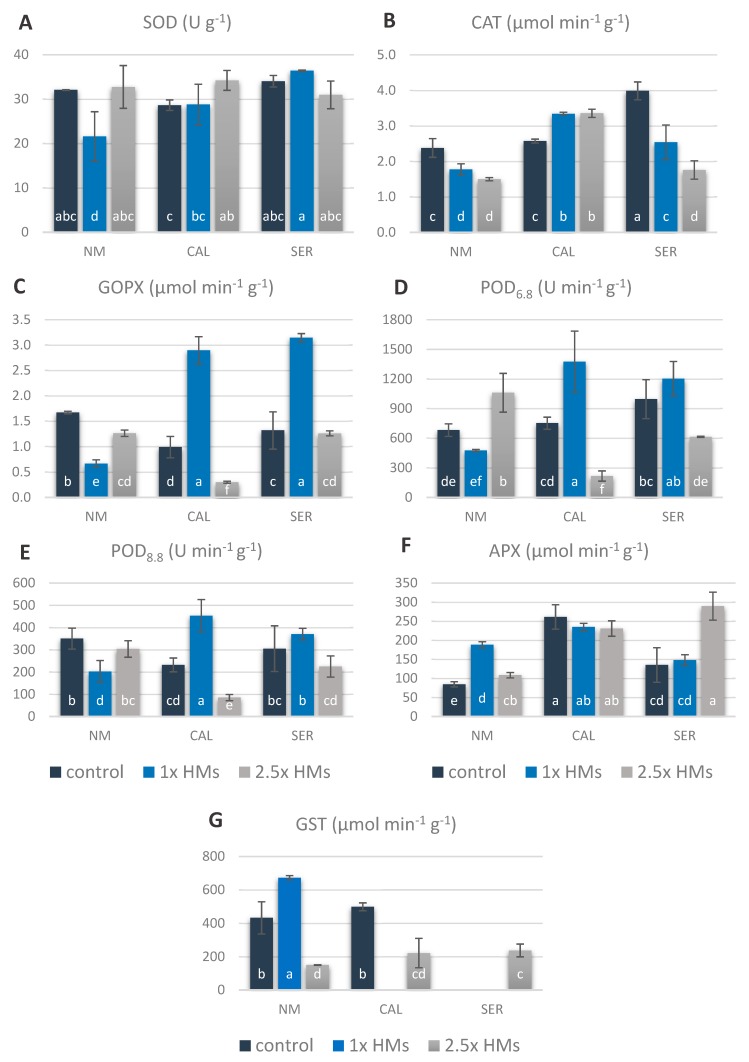
Antioxidant enzyme activity in shoots of non-metallicolous (NM), calamine (CAL), and serpentine (SER) *Silene vulgaris* ecotypes treated with heavy metals (HMs). Values are mean ± SE. Means indicated by the same letters (a–f) point to statistically significant differences according to two-way ANOVA and post-hoc Fisher’s test at *p* < 0.05. Control means the medium without the addition of Zn, Pb, Cd ions; 1× HMs—medium enriched with 714.3 μM ZnSO_4_, 3.0 μM Pb(NO_3_)_2_, and 16.4 μM CdCl_2_; 2.5× HMs means 2.5-times higher Zn, Pb, Cd concentration than the first one. Enzyme abbreviations in order of their occurrence: (**A**) SOD—superoxide dismutase, (**B**) CAT—catalase, (**C**) GOPX—guaiacol type peroxidase, (**D**) POD_6.8_—peroxidase activity at pH = 6.8, (**E**) POD_8.8_—peroxidase activity at pH = 8.8, (**F**) APX—ascorbate peroxidase, (**G**) GST—glutathione-*S-*transferase.

**Figure 6 antioxidants-09-00102-f006:**
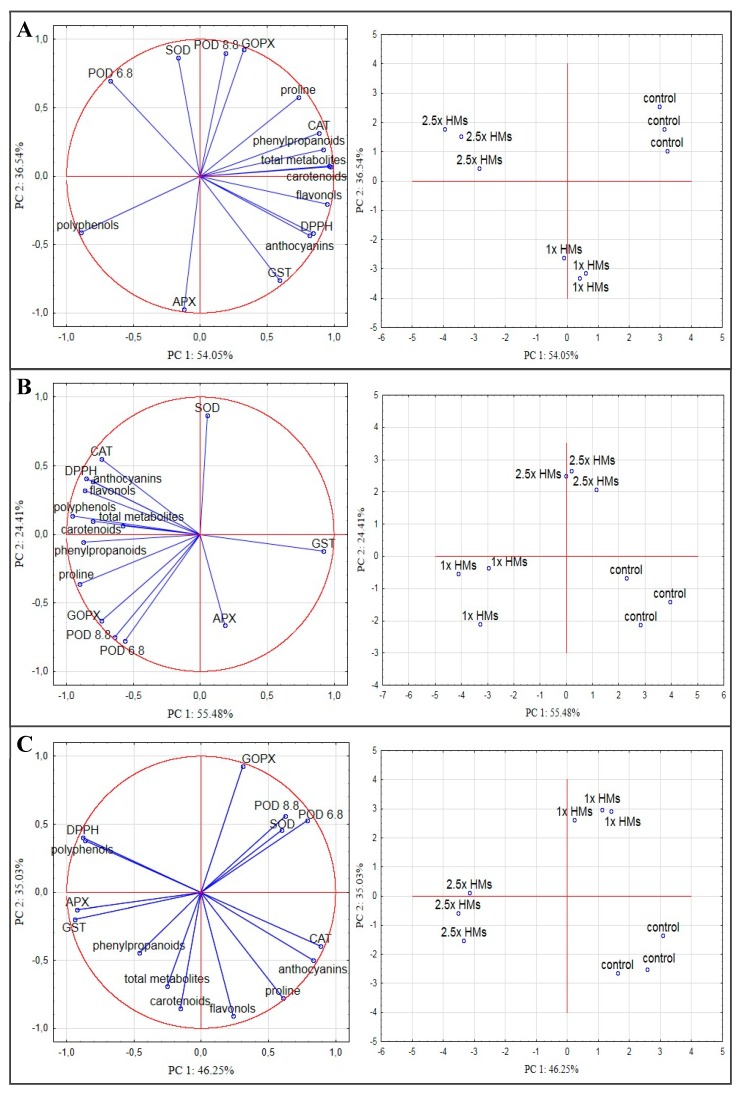
Principal components analysis (PCA) showing the relationship between examined variables (antioxidants) and different treatments for non-metallicolous (**A**), calamine (**B**), and serpentine (**C**) *Silene vulgaris* ecotypes.

**Figure 7 antioxidants-09-00102-f007:**
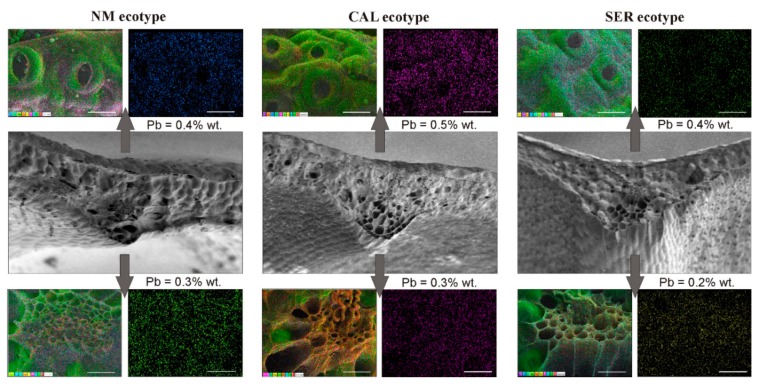
Scanning electron microscopy micrographs and elements’ localization in leaves of non-metallicolous (NM), calamine (CAL), and serpentine (SER) specimens from medium enriched with 714.3 μM ZnSO_4_, 3.0 μM Pb(NO_3_)_2_, and 16.4 μM CdCl_2_. The colored spots forming a shape of epidermis (above) or vascular bundle (below) indicate Pb presence in this tissue. Bar = 25 μm.

**Figure 8 antioxidants-09-00102-f008:**
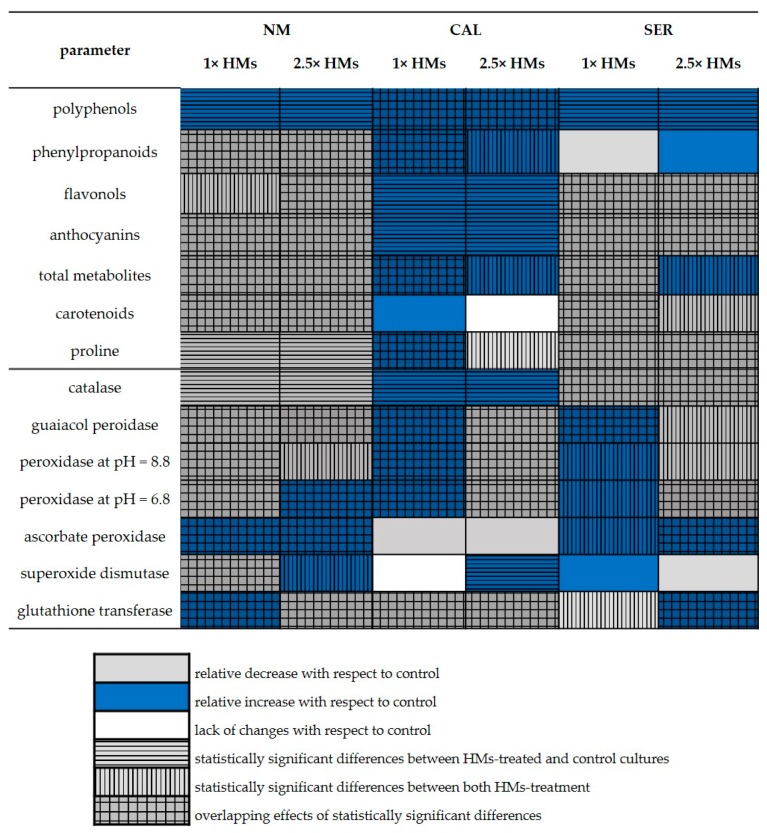
Graphical visualization of antioxidant compounds’ changes in non-metallicolous (NM), calamine (CAL), and serpentine (SER) *Silene vulgaris* shoots treated with different doses of heavy metals (HMs).

**Table 1 antioxidants-09-00102-t001:** Parameters of regenerated shoot length as well as their fresh (FW) and dry (DW) matter content in *Silene vulgaris* cultures after eight weeks of treatment with increasing concentrations of heavy metals.

Ecotype ^1^	Treatment ^2^	Minimum Length (mm)	Mean Length (mm)	Maximum Length (mm)	Median Value (mm)	Shoot FW (mg)	Shoot DW (mg)
**NM**	control	7.78	21.76 ab* ± 9.94	47.47	18.73	1223.66 a* ± 81.74	119.06 a* ± 7.95
	1× HMs	8.14	27.67 a ± 15.33	69.75	24.03	896.65 b ± 78.62	104.55 b ± 9.16
	2.5× HMs	9.02	16.98 bc ± 5.71	31.98	15.89	685.33 c ± 98.01	54.83 c ± 7.84
	5× HMs	5.17	9.30 c ± 3.66	14.11	7.71	346.67 d ± 61.46	20.81 d ± 3.69
**CAL**	control	7.70	32.78 b ± 21.12	106.01	25.43	947.00 b ± 23.51	94.87 b ± 2.36
	1× HMs	9.31	38.72 a ± 19.48	108.83	35.25	1252.67 a ± 57.55	132.91 a ± 6.11
	2.5× HMs	7.17	16.71 c ± 0.04	45.70	14.00	658.67 c ± 44.77	64.65 c ± 3.84
	5× HMs	4.58	9.88 c ± 6.05	26.90	7.11	505.13 d ± 14.11	57.33 c ± 4.51
**SER**	control	11.95	39.23 a ± 22.50	105.73	33.59	742.67 a ± 58.86	90.01 a ± 7.13
	1× HMs	7.53	26.11 b ± 13.54	76.95	22.72	458.00 b ± 25.51	47.95 b ± 2.67
	2.5× HMs	1.54	4.61 c ± 2.16	6.62	5.14	430.33 b ± 83.27	56.21 b ± 3.21
	5× HMs	n/a	n/a	n/a	n/a	n/a	n/a

^1^ NM—non-metallicolous ecotype; CAL—calamine ecotype; SER—serpentine ecotype. ^2^ control medium—without the addition of Zn, Pb, Cd ions; 1× HMs—medium enriched with 714.3 μM ZnSO_4_, 3.0 μM Pb(NO_3_)_2_, and 16.4 μM CdCl_2_; 2.5× and 5× HMs—respectively, 2.5- and 5.0-times higher Zn, Pb, Cd concentration than the first one (1× HMs). * mean values ± SD; different letters (a–d) indicate statistically significant differences at *p* < 0.05 according to one-way ANOVA and post hoc test; n/a not analyzed

**Table 2 antioxidants-09-00102-t002:** Photosynthetic pigments’ content and their ratios as well as lipid peroxidation level in the shoots of *Silene vulgaris* ecotypes cultivated for eight weeks on media enriched with different heavy metal (HMs) concentrations.

Parameter	Ecotype ^1^	Treatment ^2^		
		Control	1× HMs	2.5× HMs
Chlorophyll *a* (mg g^−1^ FW)	NM	0.436 a* ± 0.063	0.226 b ± 0.004	0.148 c ± 0.025
	CAL	0.131 c ± 0.011	0.177 bc ± 0.014	0.132 c ± 0.006
	SER	0.389 a ± 0.060	0.163 c ± 0.016	0.236 b ±0.015
Chlorophyll *b* (mg g^−1^ FW)	NM	0.121 a ± 0.024	0.082 bc ± 0.004	0.036 e ± 0.012
	CAL	0.066 cd ± 0.006	0.047 de ± 0.006	0.040 e ± 0.002
	SER	0.089 b ± 0.016	0.046 de ± 0.006	0.065 cd ± 0.003
Chlorophyll *a*/*b*	NM	3.557 cd ± 0.187	2.763 e ± 0.123	4.202 ab ± 0.636
	CAL	1.993 f ± 0.026	4.080 abc ± 0.280	3.271 de ± 0.028
	SER	4.520 a ± 0.715	3.531 cd ± 0.167	3.644 bcd ± 0.075
Chlorophyll *a+b*/carotenoids	NM	5.652 a ± 0.198	4.333 cd ± 0.094	4.594 bc ± 0.516
	CAL	3.542 de ± 0.058	3.830 cde ± 0.772	3.006 de ± 0.194
	SER	5.539 ab ± 0.812	4.296 cd ± 0.208	3.765 cde ± 0.814
MDA (μmol g^−1^ FW)	NM	37.301 e ± 0.510	56.925 c ± 1.078	73.548 a ± 2.183
	CAL	59.258 c ± 1.860	62.925 cb ± 2.541	51.065 d ± 2.495
	SER	53.054 d ± 0.746	59.247 c ± 0.614	70.946 a ± 2.177

^1^ NM—non-metallicolous ecotype; CAL—calamine ecotype; SER—serpentine ecotype. ^2^ control medium—without the addition of Zn, Pb, Cd ions; 1× HMs—medium enriched with 714.3 μM ZnSO_4_, 3.0 μM Pb(NO_3_)_2_, and 16.4 μM CdCl_2_; 2.5× HMs means 2.5-times higher Zn, Pb, Cd concentration than the first one. * mean values ± SD; different letters (a–f) indicate statistically significant differences at *p* < 0.05 according to two-way ANOVA and post hoc test.

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
