# Peer review of "Ecotype-Specific Pathways of Reactive Oxygen Species Deactivation in Facultative Metallophyte Silene vulgaris (Moench) Garcke Treated with Heavy Metals"

_antioxidants, 2020, doi:10.3390/antiox9020102_

Round 1

Reviewer 1 Report

The manuscript is generally well-written and follows the aim of the study. The subject is interesting and important for better understanding in the field of heavy metal way of action. The presented data are both quantitative and qualitative (microscopic analyses), which is a big advantage of the research.

However, some parts of the manuscript should be improved.

Numbering of the pages should be revised because some of them (2-8) are doubled and paragraph 3.2 starts with the logo page (first page of the manuscript).

All the presented data should be rounded up/down in the same way. Please, unify them.

Line 23 – “increase accumulation” or “increased accumulation”?

The expression needs unification, line 114 – “multiplication coefficient (MC)” vs. line 268 – “Micropropagation coefficient (MC)”.

Superscripts are omitted, e.g. line 225 (mM−1 cm−1).

Fig. 1 C – Lack of the letter showing statistical significance (2.5x HMs, 8 weeks).

Line 272, 423 – is “1” in “1 control medium without” necessary?

Under the Table 1:

Omit the double full stop, please.

“* mean values ± SD”, but in the table “*” should appear in three column headings: Mean length, Shoot FW and Shoot DW.

Line 323 – Please, add the space bar before “spherical structures”.

Lines 322-324 – Please, check the font size.

Line 353 – “while in SER shoots treated with HMs was similar and lower than in control one” should be clarified that both HMs doses have such a tendency. Please, improve the sentence.

Line 346 – Is it necessary to use “*” in “* mean values ± SD”?

Line 388 – Please, add the space bar before “Abbreviations”.

Lines 388-390 – Please, check the font size.

Fig. 4H – Please replace black letters with white ones.

Please, delete unnecessary space bar before superscripts, e.g. lines 407, 411, 412, 413, Fig 4-5 (g-1) and lines 480, 487 (min-1 g-1).

Figs. 4-6 – Letters above the graphs are half-visible, please, improve it.

Figs. 4-6 – Commas should be replaced by full stops in numbers, e.g. “2.5 x HMs” (Figs. 4-5), and PC1 and PC2 values (Fig. 6).

Please omit repetitions, e.g. “indicated-indicate” in “Means indicated by the same letters indicate”. It can be “point to” instead of “indicate” – lines 422, 457.

Under Fig. 4 – Please, complete the explanation “RSA”.

Line 429 – Please, add the explanation that HM supplementation did not change between doses (lack of statistical difference).

Lines 429-430 – Please, add that both HM doses elevated polyphenols content.

Line 437 – Total secondary metabolites were not discussed yet (Fig. 4G) and should not be mentioned before Fig. 4E. Please, rewrite the text.

Lines 440-443 – Please, check if it is right for CAL, and check the data with Fig. 4 (36 mg).

Line 447 – Please add, “similar to …”. Control?

Lines 448-450 – Please, check the sentences and add exceptions were necessary.

Line 460 – Please, add abbreviations for enzyme names.

Line 475 – Is it significant for POD6.8?

Lines 488-489 – Is it true for POD8.8?

Lines 492-501 – There is no data concerning GST. Please, present them in the manuscript.

Lines 507-551 – Please, check and rewrite PCA interpretation because there is some inconsistency along the text.

Line 568 – Please, add the space bar before “µm”.

Line 593 – “begging” or “beginning”?

Lines 600-601 – The sentence seems to be general to be correct. Please, specify the sentence.

Lines 699-700 – The statement that MDA remained unchanged lacks specification.

Lines 711-712; 734-736; 743-747 – Please, check if the statements are correct.

Lines 74-741 – Please, remove “however … ascertained” because it is not important from statistical point of view.

Line 762 – Please, remove “but only” because the data are statistically significant.

Line 772-774 – Lack of GST data.

Check English, e.g. line 829 – “a crucial components”.

Please, check the bibliography, e.g. line 963 – “Agric.” (capital letter); line 975 “Brassica napus” (italics); line 1016 “Nicotiana langsdorffii” (italics); line 1041 “H2O2” (subscript); line 1048 “Co2+, Ni2+ and Cd2+” (superscript)

Author Response

Please, find enclosed our responses to your comments.

Reviewer 2 Report

The submitted manuscript concerns the deactivation pathways of reactive oxigen species (ROS) in three ecotypes of Silene vulgaris treated with heavy metals. Authorss have raised the question concerning ecotype-specific mechanisms in ROS scavending and metal tolerance in three ecopypes originally derived from environment differing in metal concentration/composition. Base on in vitro cultures derived from three ecotypes and detailed biochemical and ultrastructure analysis the authors have documented that the ability of different ecotypes of the same plant species (Silene vulgaris)to grow and function properly in metal-contaminated environments is attributed to ecotype-specific antioxidant systems. The aim of the work and experimental approaches are very valuably and presented data are new. The manuscript is generally properly written and documented, therefore it requires only minor and editorial corrections (see Specific comments). Specific comments

 The conclusion are too long and should be shprtened to a few most important ones.

Abstract, line 14. The word "ecotype" befor the word "specimens" should be added.

Abstract, line 20. The statement "depending on HM concentration" should be placed in bracklets because the sentence is too long and unreadable.

The use of the term"cross-tolerance" should be clarified that it applies to "cross-tolerance to various metals".

Introduction, line 46-47. I suggest deleting the statements in brackelets, because under the influence of ROS there are more changes in lipids, proteins and nucleic acids than those listed in bracklets.

Introduction, line 53055. The sentence is too long and imprecise. I suggest shortening it to the following "All these compounds possess the ability to inactivated ROS in the complex way by their decomposition or transformation into less harmful molecules.".  

Author Response

(The authors gave the same response as above.)

Reviewer 3 Report

Dear Authors,

this work is well written and organized. The hypotheses, procedures and results are well commented. Maybe even too much.
Here are some of my minor considerations:
- Title: insert the Silene vulgaris nomenclator.
- M&M: enter the characteristics of the instrument used to read the absorbances. Line 150: enter the equation. In general, please clarify how many replicas and repetitions have been made for each analysis. This information is present in the statistical data but is not always clear. Also, were the analyzes done on the FW or DW? Line 164 and 172: FW? Please specify well.
- Results: The units of measure for MC are missing in the figures. The ANOVA letter for Serpentine ecotype at 2.5x HMs is missing. I suggest making the tables clearer. In particular with regard to the statistical analyzes conducted. At the moment it is not very easy to read these tables. Furthermore, the significance is lacking. I also suggest indicating whether the effects due to the different origins are significant. If you indicate a two-way ANOVA this is possible. In Figure 6 the codes of the images are not read.
- Discussion and Conclusions: both are very long sections. I suggest making the conclusions just a brief summary with only the main results and future prospects.

Author Response

(The authors gave the same response as above.)
